# Toxic Effects of Bt-(Cry1Ab+Vip3Aa) Maize on Storage Pest *Paralipsa gularis* (Zeller)

**DOI:** 10.3390/toxins16020092

**Published:** 2024-02-07

**Authors:** Shuang Chen, Wenhui Wang, Guodong Kang, Xianming Yang, Kongming Wu

**Affiliations:** 1College of Plant Protection, Henan Agricultural University, Zhengzhou 450002, China; c860994099@163.com; 2State Key Laboratory for Biology of Plant Diseases and Insect Pests, Institute of Plant Protection, Chinese Academy of Agricultural Sciences, Beijing 100193, China; kanggd95@163.com (G.K.); zqbxming@163.com (X.Y.); 3Institute of Insect Sciences, College of Agriculture and Biotechnology, Zhejiang University, Hangzhou 310058, China; w975480209@163.com

**Keywords:** *Paralipsa gularis* (Zeller), Cry1Ab+Vip3Aa maize, Bt insecticidal proteins

## Abstract

*Paralipsa gularis* (Zeller) is a storage pest; however, in recent years it has evolved into a considerable maize pest during the late growth stage in the border region between China and other Southeast Asian countries. Bt transgenic insect-resistant maize is an effective measure in controlling a wide range of lepidopteran pests, but there is a lack of research on the toxic effects of storage pests. We tested the toxicity of Bt-Cry1Ab, Vip3Aa, and their complex proteins against *P. gularis* via bioassay and investigated the efficiency of Bt-(Cry1Ab+Vip3Aa) maize in controlling *P. gularis* during the late growth stage of maize in the period 2022–2023. The bioassay results show that the susceptibilities of *P. gularis* to the two Bt proteins and their complex proteins were significantly different. The LC_50_ values of DBNCry1Ab (“DBN9936” event), DBNVip3Aa (“DBN9501” event), DBN Cry1Ab+Vip3Aa (“DBN3601T” event), and Syngenta Cry1Ab+Vip3Aa (“Bt11” event × “MIR162” event) were 0.038 μg/g, 0.114 μg/g, 0.110 μg/g, and 0.147 μg/g, and the GIC_50_ values were 0.014 μg/g, 0.073 μg/g, 0.027 μg/g, and 0.026 μg/g, respectively. Determination of the expression content of the insecticidal protein in different tissues of Bt-(Cry1Ab+Vip3Aa) maize shows that the total Bt protein content in different tissues was in the following order: stalk > bract > cob > kernel. However, the bioassay results show that the mortalities of *P. gularis* feeding on Bt-(Cry1Ab+Vip3Aa) maize in different tissues at different growth stages were all above 93.00%. The field trial indicates that the occurrence density of larvae and plant damage rate for conventional maize were 422.10 individuals/100 plants and 94.40%, respectively, whereas no larvae were found on Bt-(Cry1Ab+Vip3Aa) maize. In summary, this study implies that Bt-(Cry1Ab+Vip3Aa) maize has a high potential for control of *P. gularis*, providing a new technical measure for the management of the pest.

## 1. Introduction

*Paralipsa gularis* (Zeller) (Lepidoptera: Pyralidae) is native to Southeast Asia and has spread to India, Korea, Japan, northern Europe, and North America as a result of the food product trade [1,2]. It has been reported in China in the provinces of Jilin, Liaoning, Hebei, Henan, Shandong, Jiangsu, Zhejiang, Jiangxi, Sichuan, Fujian, Guizhou, and Yunnan [3]. *P. gularis* was originally recorded as a storage pest, mainly damaging stored maize, wheat, barley, soybeans, flax, dried fruits, and so on through larval feeding [4]. In recent years, due to factors such as the adjustment of planting structure and climate change, damage from *P. gularis* has shifted from storage to the field, and the degree of damage has increased year by year, posing a serious threat to the production of maize. In 2013, *P. gularis* began to occur on maize in the fields of Dehong Prefecture, Yunnan Province, China, damaging maize ears in the late growth stage, with a severe infestation rate of 20 individuals per ear and a plant damage rate of 62.0%, triggering the occurrence of ear rot [5]. In 2021, *P. gularis* appeared on maize fields in Pu’er City, Yunnan Province, China, boring the ear and stalk, with a high infestation rate of 34 individuals per ear and a plant damage rate of up to 58% [6]. In the period 2020–2022, *P. gularis* infested maize fields in Baoshan City, Yunnan Province, China, with an expansion in the area of occurrence, increase in the insect population, aggravation of the degree of infestation, advancement of the infestation fertility period, increase in the number of infested parts, and a maize yield loss rate as high as 71.7% [7]. In addition, *P. gularis* is harming maize ears in maize fields in Laos and Myanmar, adjacent to China’s Yunnan Province [5].

Chemical control, characterized by quick effects, strong emergency response, and easy operation, is one of the main means of prevention and control of pests, especially against migratory and outbreak pests [8]. In the past few decades, pesticide use in China has shown an increasing trend, and the amounts of chemical insecticides used per unit area in the period 2016–2017 was several times higher than that in Western developed countries [9]. It has been reported that since *Spodoptera frugiperda* (J. E. Smith) invaded western Yunnan Province, China, the frequency of pesticide use in the area has shown an increasing trend [10]; new high-efficiency and low-toxicity pesticides, such as spinetoram, emamectin benzoate, chlorantraniliprole, acephate, and antibiotic pesticide spinosad, have played a successful role in controlling *S. frugiperda* [11]. At present, the control of *P. gularis* is mainly achieved through the spraying of chemical pesticides at the ear stage. However, due to the larvae blooming after the milk stage, and the drilling of maize ears, the agent cannot make full contact with the insect body, and often cannot achieve good prevention and control effects. In addition, maize plants in the reproductive growth stage are tall, pesticide application is difficult, and even if pesticide is applied, it can easily cause pesticide residues [5,7]. Further, with the frequent use of chemical pesticides, most major agricultural pest populations have gradually developed resistance. For example, in Brazil, *S. frugiperda* has developed resistance to *beta*-cypermethrin with a 13-fold multiplicity of resistance [12]. In China, *Spodoptera exigua* (Hübner) with eight different geographical populations has shown moderate resistance levels to Chlorantraniliprole [13]. In addition, the extensive use of chemical pesticides can affect insect pollination, reduce the number of natural predator insects, pollute the soil and water environment, and disrupt agricultural biodiversity and the balance of ecosystems [14]. In the 1990s, Chinese farmers sprayed large quantities of chemical pesticides to control *Helicoverpa armigera* (Hübner), killing natural predators and insects, destroying the ecological balance, and eventually leading to the reemergence of *H. armigera*, which brought about a series of problems such as rapid increases in production costs, the pesticide poisoning of humans and animals, and the deterioration of the ecological environment [9]. Therefore, it is important to develop new control technologies to more effectively reduce the yield loss caused by *P. gularis* on maize and reduce the use of chemical pesticides.

Based on the above, transgenic insect-resistant maize provides an effective method by which to control lepidopteran pests. Transgenic insect-resistant maize can express proteins produced by *Bacillus thuringiensis* (Bt) with specific insecticidal effects on target pests [15], which can not only efficiently prevent and control major agricultural pests [16,17] but can also significantly reduce the use of chemical pesticides, save on production costs, and bring economic benefits; these advantages are welcomed by the majority of farmers [18,19]. In 1996, transgenic insect-resistant maize was widely planted in the United States, and by 2019, the planting area of maize with insect-resistant and herbicide-tolerant composite traits had reached 55.9 million hm^2^, accounting for 91.2% of the total planting area of transgenic maize. Dozens of countries were involved in the planting, and there were a wide variety of species, such as “Bt176”, “Bt11” and “MON810” maize expressing Cry1Ab, “TC1507” maize expressing Cry1F, “MON89034” maize expressing Cry1A.105+Cry2Ab2, “MIR162” maize expressing Vip3A, and “MON89034 × TC1507 × MIR162” maize with polygenic polymerization; these were planted in the American continent and subtropical regions [20,21]. Currently, transgenic insect-resistant maize is mainly used for the prevention and control of lepidopteran pests and coleopteran pests, and its development process can be divided into three stages: the first stage involved planting insect-resistant maize with a single Cry gene from 1996 to 2009, the second stage involved planting multiple insect-resistant maize plants with multiple Cry genes with different modes of action from 2010 to 2016, and the third stage involved planting insect-resistant maize with Cry insect-resistant genes, Vip insect-resistant genes, and RNA interference technology to control multiple target pests from 2017 to the present [20]. After more than 20 years of the development of transgenic insect-resistant maize, Bt genes mainly include genes encoding Cry and Cyt insecticidal proteins and genes encoding Vip insecticidal proteins, but Cry genes are more widely used in commercial applications [22].

Maize is one of the major grain crops in China. In 2021, China’s maize planting area was 43.3 million hm^2^, with a total output of 273 million tons [23]. Lepidopteran pests have always been one of the major factors affecting maize yields, with pest-induced yield losses accounting for 10–20% of total maize production each year [20,24]. To reduce grain loss caused by lepidopteran pests, Chinese researchers have developed several transgenic insect-resistant maize strains, and the Chinese government has successively issued corresponding safety certificates for production and application, such as DBN9936 (Cry1Ab), DBN9501 (Vip3Aa), DBN3601T (Cry1Ab+Vip3Aa), and Bt11 × MIR162 (Cry1Ab+Vip3Aa). These transgenic insect-resistant maize varieties are highly effective against lepidopteran pests, such as *S. frugiperda*, *Ostrinia furnacalis* (Guenée), and *H. armigera* [25,26,27], but their control potential against *P. gularis* remains unclear until now. Therefore, in order to develop new technology for the control of *P. gularis* and to promote the deployment of transgenic insect-resistant maize in pest management, we determined the virulence of Bt-Cry1Ab, Vip3Aa and their complex proteins against *P. gularis* and studied the insecticidal effect of Bt-(Cry1Ab+Vip3Aa) maize against *P. gularis* during 2022–2023.

## 2. Results

### 2.1. Susceptibility of P. gularis to Bt Insecticidal Proteins

The results with respect to the lethal and growth-inhibitory concentrations of four Bt insecticidal proteins, DBNCry1Ab, DBNVip3Aa, DBN Cry1Ab+Vip3Aa, and Syngenta Cry1Ab+Vip3Aa, against *P. gularis* are shown in Table 1 and Table 2. The LC_50_ values of the four Bt insecticidal proteins ranged from 0.038 to 0.147 μg/g, LC_95_ ranged from 0.606 to 0.2513 μg/g, GIC_50_ ranged from 0.014 to 0.073 μg/g, and GIC_95_ ranged from 0.072 to 0.249 μg/g. There were significant differences in the susceptibility of *P. gularis* to the four Bt insecticidal proteins, with the lowest LC_95_ and GIC_95_ being 0.276 μg/g and 0.072 μg/g, respectively, for DBNCry1Ab, and the highest LC_95_ and GIC_95_ being 2.513 μg/g and 0.249 μg/g, respectively, for DBNVip3Aa. Thus, *P. gularis* is most susceptible to Cry1Ab and least susceptible to Vip3Aa. LC_50_ values from high to low were Syngenta Cry1Ab+Vip3Aa > DBNVip3Aa > DBN Cry1Ab+Vip3Aa > DBNCry1Ab; GIC_50_ values from high to low were DBNVip3Aa > DBN Cry1Ab+Vip3Aa > Syngenta Cry1Ab+Vip3Aa > DBNCry1Ab. Both LC_95_ values and GIC_95_ values from high to low were DBNVip3Aa > Syngenta Cry1Ab+Vip3Aa > DBN Cry1Ab+Vip3Aa > DBNCry1Ab.

### 2.2. Insecticidal Protein Expression Content and Insecticidal Activity of Different Tissues in Bt-(Cry1Ab+Vip3Aa) Maize

The insecticidal protein expression content of Bt-(Cry1Ab+Vip3Aa) maize in different tissues at different growth stages is shown in Table 3. The Cry1Ab expression content was higher than the Vip3Aa expression content in all tissues of Bt-(Cry1Ab+Vip3Aa) maize. To obtain the expression of Bt-(Cry1Ab+Vip3Aa) maize insecticidal protein for different growth stages, tissues, and locations, the data for the two locations were summarized and analyzed. In general, the average expression contents of Cry1Ab, Vip3Aa, or total Bt protein in Bt-(Cry1Ab+Vip3Aa) maize were significantly different at different growth stages (*p* < 0.05), and the average expression contents of Bt insecticidal proteins decreased gradually with the maturity of maize plants (Figure 1A). The average expression contents of Cry1Ab, Vip3Aa, and total Bt protein in different tissues were also significantly different (*p* < 0.05), with the average expression contents of Cry1Ab and total Bt protein in different tissues of maize ranging from high to low as follows: stalk > bract > cob > kernel. that the average expression content of Vip3Aa ranged from high to low as follows: kernel > stalk > cob > bract (Figure 1B). The average expression contents of Cry1Ab and total Bt protein did not differ significantly between Baozang Town and Longtan Town (*p* > 0.05), while the average expression content of Vip3Aa in Longtan Town was much more than that in Baozang Town (*p* < 0.05).

The results of the laboratory tissue bioassay show that after 4 days, the mortality of the neonate larvae of *P. gularis* feeding on different tissues of Bt-(Cry1Ab+Vip3Aa) maize was as low as 93.50% and as high as 100%; these values were significantly higher than those of conventional maize, at 13.00–87.50% (*p* < 0.05) (Table 4). To clarify whether there were any differences in the mortality of *P. gularis* feeding on Bt-(Cry1Ab+Vip3Aa) maize for the growth stage, tissue, and region, the data for the two locations were summarized and analyzed. In general, there were significant differences in the average mortality of larvae feeding on R3, R5, and R6 of Bt-(Cry1Ab+Vip3Aa) maize (*p* < 0.05), with average mortality values of 98.31%, 99.38%, and 99.38%, respectively. There were also significant differences in the average mortality of larvae feeding on Bt-(Cry1Ab+Vip3Aa) maize kernel, cob, stalk, and bract (*p* < 0.05), with average mortalities of 99.42%, 99.17%, 99.33%, and 98.17%, respectively. However, there was no significant difference in the average mortalities of larvae feeding on Bt-(Cry1Ab+Vip3Aa) maize in Baozang Town and Longtan Town (*p* > 0.05), with average mortalities of 98.92% and 99.13%, respectively.

### 2.3. Control Efficiency of Bt-(Cry1Ab+Vip3Aa) Maize against P. gularis

Field surveys for two consecutive years showed that the occurrence of *P. gularis* in the summer of 2022 was more serious. The numbers of larvae per 100 plants were 208.90, 104.80, and 422.10 for conventional maize R3, R5, and R6, respectively, while no larvae were seen on Bt-(Cry1Ab+Vip3Aa) maize during the same period (*p* < 0.05). The plant damage rates were 60.30%, 75.80%, and 94.40% for conventional maize, while no damage was observed in Bt-(Cry1Ab+Vip3Aa) maize in the same period (*p* < 0.05) (Table 5). In the summer of 2023, the occurrence of *P. gularis* was relatively low, and the number of larvae per 100 plants and plant damage rates of conventional maize were 6 and 3.40%, while no *P. gularis* larvae were found on Bt-(Cry1Ab+Vip3Aa) maize. Therefore, the insect resistance effect of Bt-(Cry1Ab+Vip3Aa) maize against *P. gularis* was 100%.

## 3. Discussion

Our results show that the emerging pest of maize fields, *P. gularis*, was most susceptible to Cry1Ab, with the greatest lethality and growth inhibition rates, and least susceptible to Vip3Aa. In addition, the expression content of Cry1Ab in Bt-(Cry1Ab+Vip3Aa) maize was higher than that of Vip3Aa, and the average expression contents of Bt insecticidal proteins gradually decreased with maize growth, i.e., R3 > R5 > R6. The average expression contents of Cry1Ab and Vip3Aa in different tissues of maize from high to low were as follows: stalk > bract > cob > kernel; kernel > stalk > cob > bract, respectively. We also found that the total Bt protein expression content in different tissues of Bt-(Cry1Ab+Vip3Aa) maize plants at different growth stages were higher than the LC_95_ value of *P. gularis* throughout our study, indicating that Bt-(Cry1Ab+Vip3Aa) maize had a higher control effect. In addition, although there were differences in the content of insecticidal proteins among different tissues at different growth stages, and there were also differences in the mortality of *P. gularis* feeding on different tissues of Bt-(Cry1Ab+Vip3Aa) maize, the mortalities of *P. gularis* feeding on different tissues of Bt-(Cry1Ab+Vip3Aa) maize were all higher than 93.00%, and no occurrence and damage with respect to *P. gularis* in Bt-(Cry1Ab+Vip3Aa) maize were found in the field, suggesting that the insect resistance efficiency of Bt-(Cry1Ab+Vip3Aa) maize against *P. gularis* was 100%. Therefore, Bt-(Cry1Ab+Vip3Aa) maize could have a high potential for controlling *P. gularis*.

Because *P. gularis* has the characteristic of drilling, traditional chemical control is not very effective. However, it is similar to *O. furnacalis* in terms of biological characteristics such as spawning time and feeding damage, so the control method for *O. furnacalis* could also be used to manage *P. gularis*. Previous studies have shown that the planting of transgenic insect-resistant maize could be used to control *O. furnacalis* [26,27], and Cry1Ab is highly toxic to *O. furnacalis* [28], but Vip3Aa has weak toxic effects on *O. furnacalis* [29], which is consistent with our findings. When Bt maize expresses multiple proteins, the toxin interactions can be synergistic, antagonistic, and superimposed due to some reasons such as complementarity between the proteins, the ability of the proteins to bind to the receptor site of the insect, and the ratio of concentrations between the proteins. Synergistic effects can increase the control effectiveness. On the contrary, antagonistic effects weaken the advantages of polymerizing multiple Bt insecticidal proteins and reduce the control effectiveness [30,31,32]. Research has shown that Cry1Ia10 and Vip3Aa had different binding sites and exhibited synergistic effects in *S. frugiperda*, while in *Spodoptera eridania*, the two proteins competed for binding to the same receptor site and exhibited significant antagonistic effects [33]. When Cry1Ab and Vip3Aa19 proteins were mixed in 1:1 and 1:2 ratios by artificial diet mixing, they showed synergistic effects on *O. furnacalis*, with some antagonistic effects at the other ratios [29]. Moreover, Cry1Ab and Vip3Aa had significant synergistic effects on *S. frugiperda* [34]. In this study, the order of LC_95_ from high to low was Vip3Aa > Cry1Ab + Vip3Aa > Cry1Ab. The reason for this result may be that we used the total Bt protein (Cry1Ab + Vip3Aa) expressed by transgenic insect-resistant maize, and that *P. gularis* was the least sensitive to Vip3Aa, which in turn accounted for a certain percentage of the total Bt protein. We have not yet investigated the protein interactions between Cry1Ab and Vip3Aa on *P. gularis*, e.g., do Cry1Ab and Vip3Aa compete for binding to the same receptor site on *P. gularis*, and are both affected by concentration? In addition, the study showed that *S. frugiperda* is more sensitive to Vip3Aa protein than Cry1Ab [35], and we found that the dominant population of the pest on conventional maize was *S. frugiperda* in the local area; therefore, we do not recommend planting a single Bt-Cry1Ab maize there in consideration of the total management of maize pests. Instead, we need to further select the ratio of Cry1Ab and Vip3Aa proteins for optimal virulence against *P. gularis* and study the mechanism of their interactions against *P. gularis* to provide a scientific basis for the development of multivalent transgenic insect-resistant maize.

In addition, the resistance efficiency of transgenic insect-resistant maize against lepidopteran pests varies according to pest populations between regions [36]. Therefore, it is necessary to strengthen the population dynamics monitoring of *P. gularis*, fully grasp the occurrence and distribution of *P. gularis* on maize, and establish the susceptibility baseline of the population of *P. gularis* to Bt insecticidal proteins in different geographical regions to offer a scientific basis for evaluating the resistance management of transgenic insect-resistant maize to *P. gularis.* Previous studies have confirmed that the expression content of Bt insecticidal proteins in transgenic insect-resistant maize shows spatial and temporal dynamics with changes in the maize growth stage, the expression content of Bt insecticidal proteins gradually decreases with the maturity of maize plants, and the expression content of Bt insecticidal proteins in different tissues is also different [37], which is in agreement with our results. The expression content of Bt insecticidal proteins in transgenic insect-resistant maize is affected by factors such as maize self-regulation, temperature, sowing time, and so on, and values that are too high or too low may affect the insect-resistant effect of maize [38]. In this study, although the expression content of Bt insecticidal proteins in Bt-(Cry1Ab+Vip3Aa) maize varied depending on the growth stage, tissue, and region, all Bt-(Cry1Ab+Vip3Aa) maize exhibited strong insecticidal activity against *P. gularis.* This result indicates that the current planting of Bt-(Cry1Ab+Vip3Aa) maize is very effective in the prevention and control of *P. gularis*, which agrees with the results of previous studies [25]. However, globally, as transgenic insect-resistant maize is widely planted on a large scale, the long-term selection pressure on transgenic insect-resistant maize will inevitably lead to a series of problems in the evolution of pest resistance to transgenic insect-resistant maize, as will the application of chemical insecticides. So far, dozens of pests have been reported to be resistant to transgenic insect-resistant maize, such as *S. frugiperda* to “TC1507” maize expressing Cry1F in Puerto Rico [39,40], *Busseola fusca* (Fuller) to “MON810” maize expressing Cry1Ab in South Africa [41], *Helicoverpa zea* (Boddie) to “Bt11” maize expressing Cry1Ab [42] and “MON89034” maize expressing “Cry1A.105+Cry2Ab2” in the United States [43], *Ostrinia nubilalis* (Hubner) to “TC1507” maize expressing Cry1F in Canada [44], and so on.

The breeding of new transgenic insect-resistant maize varieties is a complex and large-scale systematic project, and obtaining an ideal trait material with outstanding target traits and genetic stability requires a large number of tests and layers of screening, during which significant manpower and material resources are required [45]. Therefore, to ensure that transgenic insect-resistant maize varieties can effectively control pests in the long term, the following points need to be addressed in production: (1) we need to monitor resistance in the application and production process of transgenic insect-resistant maize. Resistance monitoring is the basis for the management of transgenic insect-resistant maize resistance. Therefore, we should monitor the resistance development dynamics of field populations of target pests, such as *S. frugiperda*, *O. furnacalis*, and *H. armigera*, in different maize-growing areas, to recognize the early warning signs of resistance to target pests. (2) First of all, in conjunction with the transgenic planting concept of “zoning layout and source control”, existing transgenic maize varieties should be characterized, and suitable transgenic insect-resistant maize should be selected based on the types of pests occurring in different regions and their characteristics. The second strategy should involve organically combining transgenic technology with a variety of techniques like conventional breeding, molecular marker-assisted selection, haploid breeding technology, and so on to develop more and better new-generation transgenic maize varieties like multi-gene varieties. (3) Because of China’s smallholder farming pattern, farmers should assume the main responsibility for fulfilling refuge requirements when growing transgenic insect-resistant maize. The Chinese administration should popularize the science of transgenic insect-resistant maize among the general public, guide farmers to regulate the planting of transgenic insect-resistant crops, and increase the supervision of the seed quality of seed producers. In addition, under the condition of the rational implementation of a high-dose/refuge strategy, combined with integrated management techniques, such as monitoring and the interception of pest migratory populations, sexual and food baiting and light baiting, biological control, and agricultural control, we should implement a multi-dimensional, multi-level, and comprehensive control program to enhance the level of resistance management. (4) We should further strengthen cooperation with respect to the monitoring and early warning of the dynamics of pest population migrations and research on the monitoring and management of resistance to transgenic insect-resistant maize with border countries such as Laos, Myanmar, and Vietnam, and American countries such as the United States and Brazil.

## 4. Conclusions

The toxicity bioassay of Bt-Cry1Ab, Vip3Aa, and their complex proteins, as well as field trial of Bt-(Cry1Ab+Vip3Aa) maize, show that Bt-(Cry1Ab+Vip3Aa) maize has high potential for controlling *P. gularis*, providing theoretical guidance for the popularization and application of Bt maize to manage storage insect pests.

## 5. Materials and Methods

### 5.1. Collection and Rearing of P. gularis

The *P. gularis* population used in the experiment was collected from Baozang Town, Jiangcheng County, Pu’er City, Yunnan Province (22°40′55.82″ N, 101°38′51.44″ E). Adults were trapped using a vertical searching light trap equipped with a 1000 W metal–halide lamp (Modle JLZ1000BT, Shanghai Yaming Lighting Co., Shanghai, China) and brought back to the indoor area then placed in a transparent round plastic box (diameter 20 cm, height 10 cm). A cotton ball soaked in 10% honey water was placed in the bottom of the box as a food source, and a layer of medical gauze was placed on the top to facilitate the collection of adult eggs. The following day, the gauze containing the eggs was cut and placed in a 50 mL centrifuge tube until the eggs hatched, and absorbent cotton moistened with clean water was placed in the mouth of the centrifuge tube to moisturize it. Within 12 h after hatching, larvae were selected for bioassay experiments. All larvae and adults were reared in an insect rearing room at 26 ± 1 °C, 60% ± 10% relative humidity, and a 16:8 h light/dark cycle.

### 5.2. Determination of the Susceptibility of P. gularis to Bt Insecticidal Proteins

Cry1Ab and Vip3Aa and two complex proteins were derived from the leaves of Bt-Cry1Ab insect-resistant maize (“DBN9936” event), Bt-Vip3Aa insect-resistant maize (“DBN9501” event), Bt-(Cry1Ab+Vip3Aa) insect-resistant maize (“DBN3601T” event), and Bt-(Cry1Ab+Vip3Aa) insect-resistant maize (“Bt11” event × “MIR162” event), respectively. The source of the four transgenic insect-resistant maize varieties, the production method of Bt insecticidal proteins, and the expression content of different Bt insecticidal proteins were referred to in the study by Wang et al. [35].

Dilution of Bt insecticidal proteins expressed in leaves of different transgenic maize was performed with an artificial diet based on maize powders and soybean powders [46]. The concentrations of each Bt insecticidal protein dilution were as follows: 0.0957, 0.1914, 0.3827, 0.7654, and 1.5308 µg/g DBNCry1Ab; 0.0254, 0.0508, 0.1016, 0.2032, and 0.4064 µg/g DBNVip3Aa; 0.1016, 0.2032, 0.4065, 0.8129, and 1.6258 µg/g DBN Cry1Ab+Vip3Aa; 0.1393, 0.2787, 0.5574, 1.1147, and 2.2294 µg/g Syngenta Cry1Ab+Vip3Aa (µg/g: µg of Bt insecticidal protein per g of artificial diet). The appropriate level of conventional maize leaf lyophilized powder mixed with the artificial diet was used as a reference. The diluted artificial diet was evenly divided into 24-well plates (about 0.5g per well) and then 1 neonate larva was randomly accessed per well, 24 neonate larvae represented 1 replicate, and 3 replicates were set up for a total of 72 neonate larvae per concentration. Then, samples were placed in environmental conditions at 26 ± 1 °C, 60% ± 10% relative humidity, and a 16:8 h light/dark cycle. The survival of the larvae was checked after 14 days. The larvae were lightly touched with a brush, and those that could not crawl normally were regarded as dead. Each larva in control and treatment groups was weighed individually after 14 days. Then the mortality, corrected mortality, and growth inhibitory rate were calculated. In calculating the growth inhibitory rate, the idea was to weigh all larvae (dead and alive). Since the 1st instar larva was too small to weigh, and its weight was negligible compared to the weight after 14 days, we treated the weight of 1st instar larvae as 0, and the actual weight measured was the weight increase. As some larvae died at the 1st instar, we weighed the larvae that were alive. During the experiment, diets of the same composition were added or replaced on time, depending on the freshness of the diets and their consumption by feeding.

### 5.3. Determination of Insecticidal Protein Expression Content and Insecticidal Activity in Different Tissues of Bt-(Cry1Ab+Vip3Aa) Maize

Bt-(Cry1Ab+Vip3Aa) insect-resistant maize (“DBN3601T” event) and conventional maize were planted in the period May–June 2022 in Baozang Town and Longtan Town (22°47′12.62″N, 100°58′36.91″E), Pu’er City, Yunnan Province, China. All maize seeds were provided by Beijing Dabeinong Biotechnology Co., Ltd. Three fields of each variety were sown at each location, each field was larger than 667 m^2^, and the fields were 3 m apart. The planting density was 3800 plants/667 m^2^. Maize was considered to be at a particular growth stage when ≥50% of the maize plants were at the same growth stage in each field.

Field surveys showed that *P. gularis* started to damage maize ears and stalks from maize R3, so we collected samples of maize kernel, cob, stalk, and bract for R3, R5, and R6 from the field. Maize plants were randomly selected from three Bt-(Cry1Ab+Vip3Aa) maize fields and three conventional maize fields at each location, and the different tissues were individually separated into self-sealing bags, placed in a foam box with ice to be brought back indoors, and later moved to a refrigerator at −80 °C to be pre-frozen for 12 h. The pre-frozen samples were placed into a freeze dryer until completely dry and then removed (about 48 h), ground into a fine powder with a tissue grinder, mixed thoroughly, and then dispensed into 50 mL centrifuge tubes, and stored at −80 °C until use. Afterward, the samples were assayed for the expression content of Bt insecticidal proteins according to a sandwich enzyme-linked immunosorbent assay (ELISA) [35]. Measurements were repeated 3 times for each tissue.

Samples of maize kernel, cob, stalk, and bract for R3, R5, and R6 were collected from the field and brought back to the laboratory. The cob, stalk, and bract were cut into small fragments (about 2 cm) with scissors and put into 24-well plates, with 1 kernel per well for kernels and 1 fragment per well for cob, stalk, and bract. Then, 1 neonate larva was accessed in each well for determination. The 24-well plates were sealed with sealing film to prevent larvae from escaping and placed in an insect-rearing chamber at 26 ± 1 °C, 60% ± 10% relative humidity, and a 16:8 h light/dark cycle. Each tissue was connected to 50 neonate larvae as 1 replication and 4 replications were made for a total of 200 neonate larvae. According to the freshness of the maize tissues, the tissues from the same source were replaced, and the survival of the larvae was investigated after 4 days. The larvae were lightly touched with a brush, and those that could not crawl normally were regarded as dead. Then, we calculated the mortality.

### 5.4. Field Surveys of the Control Efficiency of Bt-(Cry1Ab+Vip3Aa) Maize against P. gularis

The experiment was conducted in the period May–August 2022 and 2023 in the occurrence area of *P. gularis* in Baozang Town, Jiangcheng County, Yunnan Province, China. Bt-(Cry1Ab+Vip3Aa) maize (“DBN3601T” event) and conventional maize seeds were provided by Beijing Dabeinong Biotechnology Co., Ltd. (Beijing, China). Ten fields each for Bt-(Cry1Ab+Vip3Aa) maize and conventional maize were planted per year. The area of each field was larger than 667 m^2^ and the planting density was 3800 plants/667 m^2^. Throughout the growth stages of maize, field management, such as fertilizing and watering, was carried out according to conventional methods, except that no insecticides were applied. After the emergence of maize seedlings, the W-type five-point sampling method was used to survey the occurrence of *P. gularis* larvae in Bt-(Cry1Ab+Vip3Aa) maize and conventional maize every 10–15 days. The surveys were conducted by visual inspection, with 20 plants per point, focusing on the observation of maize ears during the reproductive growth stage of maize, and the number of *P. gularis* larvae on each maize plant and the damage caused by them were recorded. Notably, because field maize plants may be damaged by some pests, such as *S. frugiperda*, *H. armigera,* and *Mythimna separata* (Walker), we only took into account the plants damaged on which we could see *P. gularis* larvae or which had typical damage characteristics of *P. gularis*.

### 5.5. Statistical Analyses

Based on the check data, the mortality of larvae in each treatment was calculated according to Equation (1), the corrected mortality was calculated according to Equation (2), and the growth inhibitory rate was calculated according to Equation (3).
Mortality = number of dead insects/total number of insects in the test × 100%(1)
Corrected mortality = (treatment mortality − control mortality)/(1 − control mortality) × 100%(2)
Growth inhibitory rate = (control weight increment − treatment weight increment)/control weight increment × 100%(3)

The susceptibility of *P. gularis* to Bt insecticidal proteins was evaluated using probit regression to generate LC_50_ (LC_95_, GIC_50_, GIC_95_) values with 95% fiducial limits (FL) for each protein. The differences among LC_50_ (LC_95_, GIC_50_, GIC_95_) values for different Bt insecticidal proteins were regarded as significant if the 95% fiducial limits of LC_50_ (LC_95_, GIC_50_, GIC_95_) values for different Bt insecticidal proteins by *P. gularis* did not overlap. A generalized linear model was used to analyze the significance of differences in the expression content of Bt insecticidal proteins in Bt-(Cry1Ab+Vip3Aa) maize and the significance of differences in mortality among *P. gularis* larvae feeding on Bt-(Cry1Ab+Vip3Aa) maize using growth stage, tissue, and location as variables. A Mann–Whitney U test was used to test the significance of differences in mortality of *P. gularis* larvae feeding on different maize varieties in respect of the same tissue at the same stage, as well as the significance of differences in larval incidence or plant damage in R3, R5, and R6 in different maize fields. All data were analyzed using SPSS 25.0 (IBM, Armonk, NY, USA).

## Figures and Tables

**Figure 1 toxins-16-00092-f001:**
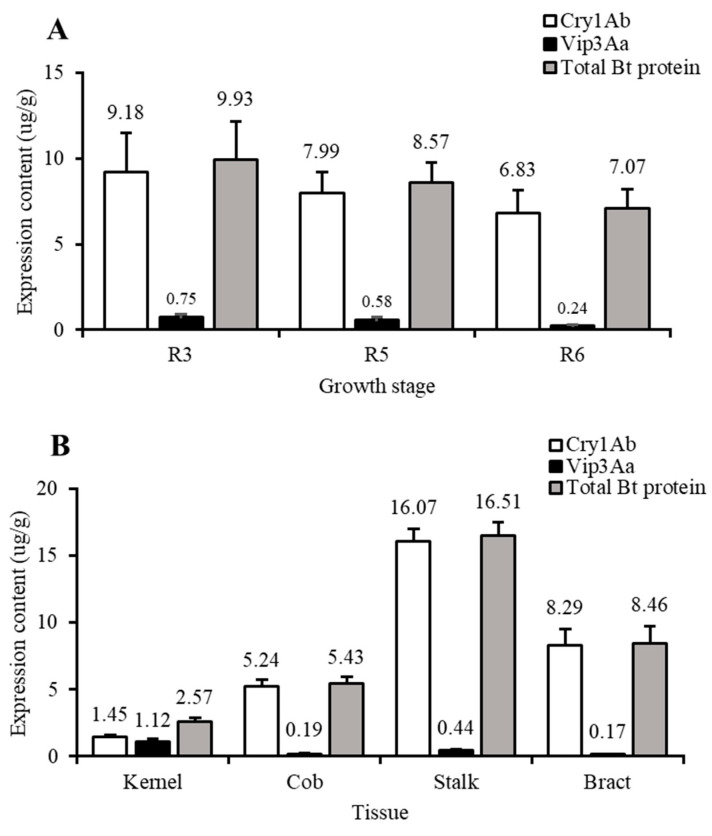
Expression content of different insecticidal proteins in Bt-(Cry1Ab+Vip3Aa) maize (“DBN3601T” event). (**A**) Different growth stages; (**B**) different tissues.

**Table 1 toxins-16-00092-t001:** Concentrations of Bt insecticidal proteins expressed in different transformation events lethal to *P. gularis* 1st instar larvae.

Bt Event	Protein	N	LC_50_ (95%FL) μg/g	LC_95_ (95%FL) μg/g	Slope ± SE	*χ* ^2^	*df*
DBN9936	DBNCry1Ab	432	0.038 (0.013–0.062) b	0.276 (0.207–0.455) c	1.916 ± 0.406	13.043	13
DBN9501	DBNVip3Aa	432	0.114 (0.088–0.150) a	2.513 (1.225–8.564) a	1.224 ± 0.169	18.141	13
DBN3601T	DBN Cry1Ab+Vip3Aa	432	0.110 (0.081–0.136) a	0.606 (0.463–0.917) b	2.217 ± 0.288	16.306	13
Bt11×MIR162	Syngenta Cry1Ab+Vip3Aa	432	0.147 (0.107–0.184) a	0.866 (0.657–1.321) ab	2.138 ± 0.277	5.096	13

N: number of insects tested. 95% FL: 95% fiducial limits. LC_50_ (LC_95_): concentration of protein (µg/g) required to kill 50% (95%) of larvae over 14 days. SE: standard errors. *χ*^2^: chi-square. *df*: degrees of freedom. Different lowercase letters in the same column indicate significant differences (non-overlapping 95% fiducial limits).

**Table 2 toxins-16-00092-t002:** Concentrations of Bt insecticidal proteins expressed in different transformation events on the growth inhibition of *P. gularis* 1st instar larvae.

Bt Event	Protein	N	GIC_50_ (95%FL) μg/g	GIC_95_ (95%FL) μg/g	Slope ± SE	*χ* ^2^	*df*
DBN9936	DBNCry1Ab	432	0.014 (0.000–0.035) b	0.072 (0.020–0.096) c	2.343 ± 0.861	4.965	13
DBN9501	DBNVip3Aa	432	0.073 (0.061–0.085) a	0.249 (0.199–0.349) a	3.088 ± 0.176	34.555	10
DBN3601T	DBN Cry1Ab+Vip3Aa	432	0.027 (0.010–0.043) b	0.129 (0.108–0.154) b	2.442 ± 0.482	3.511	13
Bt11×MIR162	Syngenta Cry1Ab+Vip3Aa	432	0.026 (0.006–0.048) b	0.160 (0.122–0.196) b	2.068 ± 0.450	1.501	13

N: number of insects tested. 95% FL: 95% fiducial limits. GIC_50_ (GIC_95_): effective concentration of protein (µg/g) required to cause 50% (95%) growth inhibition over 14 days. SE: standard errors. *χ*^2^: chi-square. *df*: degrees of freedom. Different lowercase letters in the same column indicate significant differences (non-overlapping 95% fiducial limits).

**Table 3 toxins-16-00092-t003:** Insecticidal protein expression content of Bt-(Cry1Ab+Vip3Aa) maize (“DBN3601T” event).

Growth Stage	Tissue	Cry1Ab (μg/g)		Vip3Aa (μg/g)		Total Bt Protein (μg/g)	
		Baozang Town	Longtan Town	Baozang Town	Longtan Town	Baozang Town	Longtan Town
R3	Kernel	-	1.02 ± 0.04	-	1.53 ± 0.04	-	2.55 ± 0.01
	Cob	-	6.48 ± 1.20	-	0.56 ± 0.05	-	7.04 ± 1.25
	Stalk	-	21.60 ± 0.55	-	0.64 ± 0.02	-	22.24 ± 0.56
	Bract	-	7.62 ± 0.37	-	0.27 ± 0.03	-	7.89 ± 0.35
R5	Kernel	1.66 ± 0.30	2.42 ± 0.10	0.80 ± 0.04	2.19 ± 0.05	2.46 ± 0.29	4.61 ± 0.15
	Cob	7.40 ± 0.68	4.89 ± 0.40	0.18 ± 0.03	0.06 ± 0.02	7.58 ± 0.68	4.95 ± 0.38
	Stalk	14.19 ± 2.96	15.46 ± 0.92	0.42 ± 0.03	0.74 ± 0.02	14.60 ± 2.97	16.20 ± 0.91
	Bract	15.24 ± 0.31	2.65 ± 0.18	0.20 ± 0.02	0.07 ± 0.00	15.44 ± 0.33	2.72 ± 0.18
R6	Kernel	1.12 ± 0.08	1.04 ± 0.02	0.20 ± 0.02	0.80 ± 0.03	1.40 ± 0.07	1.84 ± 0.02
	Cob	3.63 ± 0.32	3.79 ± 0.11	0.01 ± 0.00	0.15 ± 0.02	3.64 ± 0.32	3.94 ± 0.08
	Stalk	14.09 ± 1.22	14.99 ± 0.49	0.12 ± 0.01	0.29 ± 0.02	14.22 ± 1.23	15.29 ± 0.48
	Bract	7.62 ± 3.50	8.32 ± 0.87	0.02 ± 0.01	0.28 ± 0.01	7.64 ± 3.50	8.59 ± 0.87

The data in the table are averages ± standard errors. -: no samples were collected.

**Table 4 toxins-16-00092-t004:** Mortality (%) of *P. gularis* 1st instar larvae feeding on Bt-(Cry1Ab+Vip3Aa) maize (“DBN3601T” event) indoors after 4 days.

Tissue	Maize Variety	R3		R5		R6	
		Baozang Town	Longtan Town	Baozang Town	Longtan Town	Baozang Town	Longtan Town
Kernel	Bt-(Cry1Ab+Vip3Aa) maize	100.00 ± 0.00 *	99.00 ± 1.00 *	99.50 ± 0.50 *	99.00 ± 0.58 *	100.00 ± 0.00 *	99.00 ± 0.58 *
	Conventional maize	17.50 ± 1.26	20.50 ± 1.71	13.00 ± 1.29	18.50 ± 2.36	55.00 ± 3.00	27.50 ± 1.71
Cob	Bt-(Cry1Ab+Vip3Aa) maize	98.50 ± 0.96 *	98.00 ± 1.15 *	100.00 ± 0.00 *	100.00 ± 0.00 *	99.00 ± 0.58 *	99.50 ± 0.50 *
	Conventional maize	55.50 ± 3.69	19.50 ± 4.50	31.50 ± 2.63	63.50 ± 3.20	44.00 ± 3.92	29.50 ± 2.99
Stalk	Bt-(Cry1Ab+Vip3Aa) maize	99.00 ± 0.58 *	100.00 ± 0.00 *	99.50 ± 0.50 *	100.00 ± 0.00 *	98.00 ± 0.82 *	99.50 ± 0.50 *
	Conventional maize	47.00 ± 1.92	37.00 ± 5.80	41.50 ± 4.79	51.00 ± 4.12	42.50 ± 6.60	24.00 ± 4.83
Bract	Bt-(Cry1Ab+Vip3Aa) maize	93.50 ± 2.22 *	98.50 ± 0.96 *	100.00 ± 0.00 *	97.00 ± 0.58 *	100.00 ± 0.00 *	100.00 ± 0.00 *
	Conventional maize	76.00 ± 2.31	75.50 ± 1.89	85.00 ± 4.44	66.50 ± 7.63	87.50 ± 2.50	74.50 ± 3.30

The data in the table are averages ± standard errors. *: significant difference in mortality between Bt-(Cry1Ab+Vip3Aa) maize and conventional maize of the same tissue at the same growth stage (Mann–Whitney U test, *p* < 0.05).

**Table 5 toxins-16-00092-t005:** *P. gularis* damages to Bt-(Cry1Ab+Vip3Aa) maize (“DBN3601T” event) at different growth stages in the fields.

Survey Year	Maize Variety	R3		R5		R6	
		Numbers of Larvae per 100 Plants	Plant Damage Rates (%)	Numbers of Larvae per 100 Plants	Plant Damage Rates (%)	Numbers of Larvae per 100 Plants	Plant Damage Rates (%)
2022 year	Bt-(Cry1Ab+Vip3Aa) maize	0.00 ± 0.00	0.00 ± 0.00	0.00 ± 0.00	0.00 ± 0.00	0.00 ± 0.00	0.00 ± 0.00
	Conventional maize	208.90 ± 27.53 *	60.30 ± 5.18 *	104.80 ± 13.06 *	75.80 ± 5.60 *	422.10 ± 44.70 *	94.40 ± 3.22 *
2023 year	Bt-(Cry1Ab+Vip3Aa) maize	0.00 ± 0.00	0.00 ± 0.00	0.00 ± 0.00	0.00 ± 0.00	0.00 ± 0.00	0.00 ± 0.00
	Conventional maize	6.00 ± 2.15 *	3.40 ± 0.73 *	0.80 ± 0.61	0.4 ± 0.27	1.40 ± 0.99	0.80 ± 0.44

Data in the table are averages ± standard errors. *: significant differences in the number of larvae per 100 plants or plant damage rates between Bt-(Cry1Ab+Vip3Aa) maize and conventional maize in the same growth stage (Mann–Whitney U test, *p* < 0.05).

## Data Availability

Data are contained within the article.

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
