# Peer review of "Toxic Effects of Bt-(Cry1Ab+Vip3Aa) Maize on Storage Pest Paralipsa gularis (Zeller)"

_toxins, 2024, doi:10.3390/toxins16020092_

Round 1
Reviewer 1 Report
Comments and Suggestions for Authors
The manuscript describes the effects of the bacterial insecticidal toxin combination (Cry1Ab and Vip3Aa) in maize on Paralipsa gularis larvae. I believe that this is the first report of these toxin combination against P. gularis larvae.
Table 1: Authors use the maize producing Cry1Ab along, Vip3Aa alone, and two different varieties producing both Cry1Ab and Vip3Aa. Cry1Ab is far more effective than Vip3Aa, and surprisingly, than Cry1Ab+Vip3Aa. It almost seems that these two toxin proteins work against each other and there is no synergism brtween them. This reviewer would like to see more discussion on this.
Table 3: Authors mention that samples collected for R4, R5 and R6 (line 405) in Materials and Methods. Therefore, if there are no data collected for R3 in Baozang town, this reviewer suggests authors delete R3 data and replace them with R4 data for both towns like R5 and R6. Also, when comparing the data presented in Table 3 and Figure 1, this reviewer was not clear how the numbers in Figure 1 generated. Were the two towns' data in Table 3 combined and averaged out? Even so, the numbers between Table 3 and Figure 1 do not match. If possible, it is the best to present using only one format (either a table or a bar graph).
Table 4: If the data presented in Figure 1 are correct, Cry1Ab is not produced much in kernel compared to cob, stalk or bract. Yet P. gularis larval mortalities are almost 100% regardless of stages. Considering Vip3Aa production in these four tissues are almost none (Figure 1B), this reviewer wonders how this happened.
Comments on the Quality of English LanguageThis reviewer think that English in the manuscript is fine.
Reviewer 2 Report
Comments and Suggestions for Authors
L118. replace "but" with "But"
L118-119: Any prior evaluation of transgenic maize resistance strains control potential against P. gularis? I would suggest revise the following sentence "but resistance to P. gularis has not been evaluated"
Results:
Table 1. Mention instar used
Table 2. In the table legend, please indicate how the growth inhibitory concentrations are defined.
Fig. 1= What bars and verticle lines in these bars are indicating?
Table 4= 1st instar? Please indicate the instar. The table should be detailed enough to stand alone. It is not clear data in this table are taken from lab experiments or fields. Please move the % after mortality.
Table 5: Please mention % with plant damage rate
Reviewer 3 Report
Comments and Suggestions for Authors
This manuscript demonstrated the toxicity of Cry1Ab and Vip3Aa produced from transgenic maize against Paralipsa gularis larvae. The results are convicing and could be a basis for further development. Following points should be considered.
1. Vip3Aa expression level was very low compared to Cry1Ab in the DBN3601T (Table 3). Therefore, the insecticidal activity would mainly come from Cry1Ab. However, results in Table 1 showed significant lethal concentrations between Cry1Ab alone and Cry1Ab+Vip3Aa. Is it possible that Cry1Ab and Vip3Aa are antagonists? This point should be discussed clearly to make sure that putting more toxin genes into the transgenic plant does not lower insecticidal activity.
2. Section 2.3 Control efficiency (lines 200-210). Field surveys reported only P. gularis. Are there any other insects in the field? Normally, the control group (conventional maize) can be damaged by many insects. This fact should be mentioned in the results and discussion.
3. Discussion section (lines 285-343). Authors discussed a lot about high dose and refuge strategy which is out of the scope of this manuscript. Insect resistant management, guidance and cooperation policy are also out of scope. This part is not necessary.
4. Conclusion section (lines 350-352). Authors claimed that Bt-(Cry1Ab+Vip3Aa) maize is an effective method for controlling P. gularis. This conclusion is not supported by the results. Table 1 clearly shows that Bt-DBNCry1Ab exhibited the highest activity. Therefore, the Bt-maze expressing Cry1Ab alone should be evaluated comparing to Bt-maize expressing Cry1Ab+Vip3Aa (Tables 3-5).
5. Section 5.2 determination of the susceptibility (lines 376-380). What is the composition of the artificial diet? The toxin concentration used in this experiment was quite different between Cry1Ab and Vip3Aa. Why did they use Vip3Aa much lower than Cry1Ab? Moreover, the LC50 of Cry1Ab shown in table 1 is 0.038 microgram/ml that falls outside the lowest concentration used experimentally. This LC50 value might be too extrapolated.
6. Growth inhibitory rate (lines 391-392). It is not clear whether all larvae were weighed (both dead and alive) or only the live larvae. Body mass increase was used for GIC calculation (lines 450-451). This means that each insect must be weighed before and after completing the experiment. Since the neonates were used in this experiment, it is quite difficult to weigh such a tiny insect at the beginning of the test. How can they determine the neonate weight?
Round 2
Reviewer 1 Report
Comments and Suggestions for Authors
Authors address all of the reviewer's comments in the revision.
Reviewer 3 Report
Comments and Suggestions for Authors
Authors have made substancial changes to the manuscript. All comments have been addressed positively.